# Position: Peer Review in ML/AI Conferences Should Separate Publication from Presentation and Offer Non-Anonymous Review Tracks

**Nihar B. Shah** [1]

## Abstract

In this position paper, we enumerate a number of problems with the current peer-review process based on extensive empirical evidence. We argue for two structural reforms: (1) separating publication from presentation via a four-step process that first evaluates soundness, publishes all sound papers, then uses community-based ratings to select presentations; and (2) offering parallel anonymous and non-anonymous review tracks, where the non-anonymous track releases all review data publicly to increase accountability and generate valuable research datasets. We argue how our proposed policies can mitigate these problems. We urge the community to leverage the learnings from the experiments conducted in peer-review processes and incorporate evidence-based policy design.

## 1. Introduction

Peer review has two fundamental evaluative objectives (Benos et al., 2007; Jefferson et al., 2002)

1. **Ensuring soundness**: Verifying that published research is technically sound, with the claims made in the paper being supported by appropriate evidence (e.g., correct proofs, valid experimental methodology, justified narrative).

2. **Filtering for quality**: Selecting research that is more interesting, novel, or potentially impactful from among the pool of technically sound submissions.

In this position paper, we take an evidence-based approach to understand the challenges in the current peer-review processes, from the lens of these evaluative objectives. We

[1]Machine Learning and Computer Science Departments, Carnegie Mellon University. Correspondence to: Nihar B. Shah <nihars@cs.cmu.edu>.

*Proceedings of the 43rd International Conference on Machine Learning*, Seoul, South Korea. PMLR 306, 2026. Copyright 2026 by the author(s).

briefly outline the challenges (Shah, 2022) in this paragraph, and detail them further with supporting evidence in later sections. First, a major challenge is that of the review process missing key flaws in the paper. For instance, in one controlled experiment, only 1 out of 79 reviews caught deliberately-inserted flaws. Second, when filtering for quality, decisions remain highly inconsistent, and additionally, review scores fail to predict future impact. The accuracy of the process is further compromised by multi-criterion evaluation, which is known to dilute evaluation quality, and the conflation of publication and presentation goals, which forces distinct author needs into a single pipeline. Further, the widespread practice of resubmissions wastes significant resources by unnecessarily multiplying the reviewer burden and overloading them. These issues are further compounded by a lack of accountability for review quality and absence of suitable incentives for quality reviewing. There is also an absence of critical publicly available peer-review data, hampering research on this topic. Finally, we are in the midst of a seismic change driven by AI, leading to an increase in number of submissions, but AI also serves as a tool for evaluation at least for correctness (Shah, 2022, Section 11.3).

As a consequence of the aforementioned problems, we have ended up with a system that all invested parties complain about. Authors are unhappy due to seemingly arbitrary decisions, while the stakes are very high for them in terms of their career progressions. Diligent reviewers feel their efforts go unappreciated. The scientific record is polluted with both false positives (flawed papers that slip through) and false negatives (good papers that are rejected due to noise).

In this position paper, we propose concrete policies that we argue mitigate these failures: (1) a four-step review process that separates publication from presentation, evaluating soundness first and using community ratings to select presentations; and (2) parallel anonymous and non-anonymous review tracks that increase accountability and generate public datasets for peer review research. Either reform can be adopted without the other, though we believe both would benefit the community.

We now present each reform in turn, first documenting

the specific problems it addresses with empirical evidence, then describing the proposed solution and how it resolves those problems.

## 2. Reform 1: Separate Publication from Presentation

### 2.1. Problems In Current Review Processes

#### 2.1.1. FAILURE TO ENSURE SOUNDNESS

A fundamental objective of peer review is to ensure the technical soundness of published research. The evidence suggests that the current ML/AI conference review process breaks in the pursuit of this objective.

The most direct evidence for our community comes from a controlled experiment conducted in the review process of a premier AI/ML conference during 2022–2024 (Shah, 2022, Section 10.1.2). Here, the experimenters created a paper with three variants, and in each version, included one critical error in a claimed key contribution. These errors appeared in the main text of the paper and there were no appendices. The program chairs (who were collaborating in this experiment) then obtained a total of 79 reviews in the conference. An analysis of these reviews found that:

- Only 1 out of 79 reviews questioned the erroneous claim of the paper. Additionally, 19 reviews said the erroneous part was sound, 6 reviews said it was straightforward (one said both straightforward and sound), and 54 made no comment on the erroneous technical part.

- Reviews were annotated along various criteria of quality. Review quality showed no correlation with self-reported confidence (Kendall's $\tau_b = -0.108$, $p = 0.98$) and self-reported expertise (Kendall's $\tau_b = -0.002$, $p = 0.30$).

- Very few reviews contained substantive comments on methodology. Reviewers tended to focus on the introduction and formulation sections or the experimental results section, while hardly making any substantive comments about the core methods.

While this experiment has limitations in terms of generalizability (it involved one paper with specific types of errors), it provides the most direct evidence we have about error detection in ML/AI conference peer review. Note that these outcomes are worse than patterns observed in other scientific fields: see Table 1.

Problems have also been identified even in papers receiving best paper awards at ICML 2022 and 2023 (Carlini et al., 2022; Orabona, 2023). Given that the most scrutinized papers contain critical claims that are not backed up

*Table 1.* Error detection rates in controlled experiments outside of computer science. Also note that an additional analysis of the data from Schroter et al. (2008), conducted in Shah (2022), found that over 90% of reviewers caught at least one error (they may have stopped evaluating further after catching one).

| Study | Errors | Reviews | Detected |
|---|---|---|---|
| Baxt et al. (1998) | 10 | 203 | 34% |
| Godlee et al. (1998) | 8 | 221 | 25% |
| Schroter et al. (2004) | 9 | 1,380 | 31% |
| Schroter et al. (2008) | 9 | 1,390 | 31% |
| Emerson et al. (2010) | 5 | 210 | 8% and 17% |

with clear evidence, it is not surprising that the evaluation of methodological soundness is even weaker in the regular peer-review process.

One might hope that post-publication review could catch errors that slip through the initial review process, and that the scientific record will efficiently correct itself. However, this mechanism has proven insufficient. Post-publication review is sporadic and unsystematic, and depends on readers choosing to scrutinize published work and then taking the effort to publicly report issues. The errors in the aforementioned ICML best paper awardees were identified only because prominent researchers happened to examine them closely due to the awards. Despite these public criticisms, the papers have continued to accrue over a hundred citations, many of which were highly influenced by these papers. Furthermore, even when errors are identified post-publication, the academic incentive structure provides little reward for this labor-intensive work (Tennant et al., 2017). Further exacerbating these issues is the near absence of any policies for corrections or retractions in ML/AI conference proceedings.

#### 2.1.2. FAILURE TO FILTER RELIABLY

Let us now look at the second objective of peer review – that of selecting more interesting or impactful research. It is important to note that currently the decisions on these subjective opinions are made using only a handful (usually two to four) of opinions.

**Large Inconsistencies Across Reviewer Panels.** The famous NeurIPS 2014 consistency experiment (Lawrence & Cortes, 2014) randomly assigned submissions to two independent program committees. They found that 57% of papers accepted by one committee were rejected by the other. Similar results emerged in a 2021 replication (Beygelzimer et al., 2021).

Three similar experiments have been conducted in the realm of reviewing of research proposals (Obrecht et al., 2007; Fogelholm et al., 2012; Pier et al., 2018). In each of

these cases, the agreement in the final decisions across the two panels was low.

**No/weak Correlation with Impact.** If peer review were effective at identifying important work, we would expect review scores to predict future impact. We now discuss various studies that consider several ways of measuring impact. Ragone et al. (2013); Weitzner et al. (2024) find that the reviewer scores are nearly uncorrelated with the number of subsequent citations for accepted papers, and Connolly et al. (2014) find reviewer ratings to be uncorrelated with citations or downloads. Eysenbach (2022) finds reviewers' ratings of perceived impact to be uncorrelated with citations, but sometimes correlated better with altmetrics like social media impressions. A retrospective analysis of the NeurIPS 2014 conference finds no significant correlation of the review scores for accepted papers with subsequent citations (Cortes & Lawrence, 2021): *"If we accept that final paper citation counts are some measure of paper quality, then we see that reviewers fail to capture this in their scores."* Cortes & Lawrence (2021) find a weak correlation between the reviewer scores and future citations, although it is challenging interpret such a result on the rejected papers due to the differing paths authors could have taken for the papers after rejection. Patel et al. (2024) find no significant correlation of reviewer rating scores with either two-year citations or altmetrics. Interestingly, Patel et al. (2024) also do not find any significant difference in two-year citations between accepted and rejected manuscripts, although the altmetric scores are significantly higher for accepted papers. Finally, Schroter et al. (2022) explicitly ask evaluators to forecast future citations, and find that the evaluators fail to make such predictions accurately.

### 2.1.3. RESUBMISSIONS INCREASE THE REVIEW LOAD

A large proportion of papers are repeatedly resubmitted across conferences. As the IJCAI program chairs noted, there is a *"large proportion of papers which are repeatedly re-submitted from one conference to the other"* (Bessiere, 2021). More concretely, at the 2017 IEEE S&P conference, 40% of submissions were resubmissions. At NeurIPS 2014, of 1,264 rejected submissions that could be traced, 427 were later published at other top venues (Cortes & Lawrence, 2021).

Reviewers are already overloaded with reviewing duties. Papers are frequently rejected at conferences for various reasons including subjective opinions on interestingness, and are then resubmitted elsewhere. This recycling increases aggregate reviewer burden as the same papers must now be reviewed multiple times.

### 2.1.4. CURRENT APPROACHES INVOLVING MULTI-CRITERION EVALUATION REDUCE ACCURACY

Reviewers in current ML/AI conferences are tasked with simultaneous multi-criterion evaluation, where they must evaluate both soundness and interestingness. As we discuss below, simultaneous multi-criterion evaluation suffers from problems.

A controlled experiment by Lane et al. (2024) tested whether evaluation accuracy differs when reviewers judge several criteria at once versus focusing on one at a time. Reviewers of research proposals were randomly assigned either to evaluate multiple criteria simultaneously, or to assess a single criterion. Further, the researchers could validate the single criterion's accuracy, thereby also allowing for a direct comparison of accuracy across conditions. They found that reviewers who focused on one criterion provided more critical and informative feedback, probing more deeply and relying less on broad impressions. In contrast, reviewers evaluating multiple criteria at once gave less nuanced assessments and tended to blur distinct attributes.

### 2.1.5. PUBLICATION AND PRESENTATION ARE CONFLATED

Different authors have different goals from the conference peer-review process. Some authors look for exposure of their work via presentation at conferences. But some others are primarily (and often solely) interested in a publication credential. After all, publication at a top-tier conference itself can significantly boost career prospects, both in academia and the industry. An informal data-collection exercise at the ICML 2025 conference conducted by the author (Shah, 2025, Slide 24) found that of 205 poster sessions observed, 93 had both poster and presenter, 80 had only the poster, and 32 had neither. Along these lines, when conferences were virtual during COVID, a non-negligible fraction of authors did not attend the online sessions to present.

### 2.1.6. MISMATCH BETWEEN INSTITUTIONAL EXPECTATIONS AND PEER-REVIEW REALITY

Publication in "top-tier" ML/AI conferences is considered highly prestigious and is given a heavy weight in hiring and promotion processes. For instance, China's academic evaluation system has historically classified venues like NeurIPS, ICML, and ICLR as "A1" (top-tier) publications that carry significant merit in hiring, promotion, and funding decisions (China Computer Federation). Major technology companies emphasize publications at top ML venues in their research hiring. For example, Google Research job postings (Google, 2025) for research scientists

list as preferred qualifications: "Publications in top machine learning conferences."

The institutions governing hiring and promotions often consider peer review as a "stamp of quality" (Tennant et al., 2017) on the research, with presumptions that a set of experienced researchers with deep expertise have found the work interesting and impactful. However, in reality, the decisions based on highly subjective opinions about the interestingness or predicted potential impact of a work are made via only a small number (two to four) of opinions. Furthermore, among these decision makers, a large fraction of reviewers in ML/AI conferences are early-career researchers, including those in their first or second year of PhD, as well as masters students and occasionally undergraduates (Shah et al., 2018; Stelmakh et al., 2021). This creates a significant gap between institutional expectations of experienced opinions and the reality of how reviewing is actually conducted.

### 2.1.7. MISMATCH BETWEEN PUBLIC PERCEPTION AND PEER-REVIEW REALITY

Popular news articles about scientific research frequently distinguish between peer-reviewed and non-peer-reviewed papers (e.g., see Whang, 2023). Here, peer review is stated as a stamp of soundness, and this also reflects the public perception regarding peer review. However, current review processes belie this perception in two ways. First, as discussed previously, there is lack of emphasis on soundness. And second, when papers are rejected based on subjective interestingness criteria, they are also prevented from obtaining the soundness stamp.

### 2.1.8. PROBLEM: COLLUSION RINGS AND IDENTITY THEFT

Collusion rings involve researchers making pacts with each other: a researcher in the collusion ring will accept the other's papers (or even nominate for the best paper award) in exchange for some favor (Vijaykumar, 2020; Littman, 2021). Identity theft involves a researcher signing up to review pretending to be someone else, often using verified email addresses from the impersonated researcher's institution (Shah et al., 2025). These issues of fraudulent behavior unfortunately exist and are a major pain-point for our conferences.

### 2.2. The Proposed Reform: A Four-Step Process

We propose restructuring the conference review process into four distinct steps as described below.

**Step 1: Soundness Evaluation.** The first step in the proposed evaluation scheme will evaluate submissions for technical soundness and basic topical relevance, using cri-

teria similar to those employed by the Transactions on Machine Learning Research (TMLR) journal (TMLR, 2024). TMLR's acceptance criteria focus on whether claims are well-supported by evidence, and whether it has some relevance to the audience. It explicitly excludes subjective assessments of "importance" or "impact". Reviewers in this step address a narrow set of questions: Are the claims supported by the provided evidence? Are the proofs correct? Is the experimental methodology valid? Is the narrative made in the paper justified? This focused evaluation allows reviewers to invest their limited attention on the task that matters most. Note that while criteria such as novelty are not explicitly used to make the decisions, they do factor in the TMLR criteria – for instance, a paper claiming novelty on something that has been done before will not meet the criterion because the claim of novelty is not substantiated with appropriate evidence.

The evaluation in this stage can *combine human and AI efforts*. Recent research has begun exploring the use of LLMs to assist with peer review tasks. Studies have shown that LLMs can identify certain types of errors in scientific papers, including mathematical inconsistencies and logical flaws (Liu & Shah, 2023; Xi et al., 2025). For instance, for the experiment discussed in Section 2.1.1 involving papers with deliberately inserted flaws and 79 reviews, GPT-4 (which was the frontier model before the experimental papers were released publicly) consistently identified one of the three errors, and identified a second one with steering (Shah, 2022, 11.3.2). Another study (Liu & Shah, 2023) found that GPT-4 could detect about 50% of the errors in short papers. A study by Xi et al. (2025) find that among hard errors, frontier models can flag 39% of them. Thus, instead of using LLMs to replicate scores from past conferences (such as ICLR which as publicly available reviews) — an endeavor which faces a number of problems – LLMs can be more useful in objective tasks such as checking soundness (Shah, 2025).

Step 1 may also include a rebuttal stage, allowing authors to rebut reviewer criticisms and/or revise their paper. Papers that pass Step 1 proceed to Step 2 below. Papers that fail Step 1 receive feedback identifying the specific issues. Unlike the current system where rejection could simply be due to a reviewer claiming to not find the paper interesting, rejection at this step carries actionable information on where the claims made in the paper are not met with appropriate evidence, or why the paper falls outside the relevance spectrum of the conference.

**Step 2: Publication.** All papers that pass the soundness evaluation are accepted for publication in the conference proceedings. Withdrawals are no longer permitted after Step 1. Any withdrawal would be treated as a retraction (Retraction Watch, 2024). This ensures that techni-

cally sound work quickly enters the scientific record regardless of subjective assessments of interestingness.

This step represents the most significant departure from current practice. Under our proposal, a publication in Step 1 certifies that the work is reasonably checked to be technically sound (i.e., the claims made in the paper are supported by appropriate evidence) and topically relevant. This is a signal with a clear meaning that institutions and hiring committees can use, unlike the current signal that conflates soundness with the subjective preferences of three anonymous (and often inexperienced) reviewers. One would treat papers at this stage as a publication on arXiv which has undergone additional scrutiny for soundness.

**Step 3: Community Excitingness Rating.** A primary purpose of an in-person conference is to facilitate scientific exchange—attendees come to learn about exciting new developments, engage with novel ideas, and interact with researchers working on topics of mutual interest. This purpose should guide how we select which papers to present.

After publication decisions, the community provides ratings on which papers they would most like to see presented. This step assigns more reviewers in addition to those who already read the paper in step 1. Given that the paper has already been vetted for soundness, the new reviewers need not read the paper in detail. Based on a more perfunctory reading, they provide their opinions on subjective aspects such as excitingness and potential impact, essentially answering the question "What do you want to attend at the conference?". Given the lower per-paper effort, *many more reviewers are assigned to each paper in this step*.

We now discuss the assignment of reviewers to papers for this step. One approach is to directly extend current reviewing systems: assign reviewers to papers in the same manner as done in the current peer-review process (Shah, 2022, Section 3), but with more reviewers per paper and more papers per reviewer. The reviewers assigned to a paper can also include those assigned in Step 1, since they have already read the paper. Note that the assigned reviewers thus are also qualified experts, drawn from the same pool as in current conferences. This would be our recommended approach to start out. That said, going ahead conference could explore other approaches. For instance, registered conference attendees could vote on papers they find most interesting; a weighted voting system could give more weight to researchers with demonstrated expertise in relevant areas; or a prediction market mechanism could aggregate community beliefs about which papers will prove most influential.

**Step 4: Presentation Selection.** Based on community ratings, a subset of papers is selected for in person presentation at the conference. The authors of the remaining papers have an option to make available a video presentation on the conference website, ensuring that all published work has an opportunity for dissemination.

This step preserves the valuable function of conferences as venues for scientific exchange while decoupling it from the credentialing function of publication. Authors who seek visibility compete for presentation slots; authors who primarily need a publication record have already achieved their goal in Step 2.

### 2.3. How This Reform Addresses the Problems

**Problem: Failure to ensure soundness.**
*How it addresses this problem:* Step 1 focuses exclusively on soundness, allowing reviewers to concentrate on technical soundness without being distracted by subjective judgments about excitement. As noted in past research described earlier, single-criterion evaluation improves accuracy, as reviewers dig deeper when not simultaneously judging novelty and significance.

**Problem: Failure to filter reliably.**
*How it addresses this problem:* Community rating in Step 3 aggregates many more opinions than 3 reviewers, reducing noise in excitingness assessments. The inherently subjective question of interest is handled separately from soundness, with broader input.

**Problem: Resubmissions increase reviewing load.**
*How it addresses this problem:* Once a paper passes soundness review, it is published. Repeated submissions of rejected papers to different venues diminish substantially, as technically sound work already enters the scientific record.

**Problem: Multi-criterion evaluation reduces accuracy.**
*How it addresses this problem:* Reviewers in Step 1 focus only on soundness (along with basic topical relevance checks). Reviewers in Step 3 then focus on subjective aspects. The separation of soundness evaluation from excitingness rating means each task receives focused attention.

**Problem: Publication and presentation are conflated.**
*How it addresses this problem:* Authors who want credentials get them through publication, while authors who want exposure compete for presentation slots. Each group can get what they need. Additionally, with low probability of presentation for unexciting work, the marginal submission becomes less attractive, potentially reducing overall submission volume.

**Problem: Mismatch Between Institutional Expectations and Peer-Review Reality**

*How it addresses this problem:* In the proposed reform, there is vetting of published research in terms of its soundness in the first step, and then selection of interestingness based on a much broader opinion. These aspects are closer to what hiring and promotion policymakers expect, as compared to the current peer-review and publication processes.

**Problem: Mismatch Between Public Perception and Peer-Review Reality**

*How it addresses this problem:* Emphasis on soundness addresses the first problem stated earlier. The publication of the paper after Step 1 (independent of the outcomes of Step 3) ensures that even if the paper is not considered interesting enough, it still gets a public stamp of having been vetted for soundness.

**Problem: Collusion Rings and Identity Theft**

*How it addresses this problem:* The proposed structure can mitigate these issues in two ways. First, the publication in Step 1 is simply based on soundness and does not carry a high degree of prestige, unlike the (arguably inflated) prestige attributed to acceptances in current conferences. Thus there is less incentive to game. The review process in Step 3 is now based on assigning many more than three reviewers to each paper, thereby reducing the chances that a few colluding reviewers or a fake reviewer can significantly alter the outcome of a paper.

## 3. Reform 2: Offer Anonymous and Non-Anonymous Tracks

### 3.1. Problems In Current Review Processes

#### 3.1.1. NO INCENTIVES FOR HIGH-QUALITY REVIEWING

The current system provides little reward for excellent reviewing. A reviewer might spend hours uncovering a subtle but fatal flaw in a proof, or provide detailed feedback that fundamentally reshapes a paper. However, these contributions under the current system would remain largely invisible to the community. While "outstanding reviewer" awards exist, they reveal nothing about what the reviewer actually did; the community never learns which challenging errors were caught or which constructive suggestions transformed a weak submission into a strong one. Furthermore, these awards are opaque in terms of their criteria, thereby further diluting their importance. Thus the individual incentive to provide high-quality reviews is weak.

#### 3.1.2. NO ACCOUNTABILITY FOR POOR REVIEWING

Conversely, reviewers face no consequences for low-effort or unfair reviews. The anonymity of the current system protects reviewers from accountability for their assessments.

A persistent problem in peer review is the prevalence of vitriolic, dismissive, or unnecessarily harsh reviews. Survey research has documented that a substantial fraction of researchers report receiving reviews they perceived as unfair, rude, or personally attacking (Silbiger & Stubler, 2019; Gerwing et al., 2020). The anonymity of the current system, while protecting reviewers from retaliation, also shields them from any social consequences for unconstructive behavior. Reviewers can dismiss work with one-line rejections, make ad hominem criticisms, or express frustration in ways they would never do in a face-to-face interaction. This toxicity not only harms individual authors but also degrades the collaborative spirit that should characterize scientific discourse (Tennant et al., 2017).

#### 3.1.3. LACK OF RECIPROCITY

The reviewing system depends on community members contributing reviews, but this contribution is decoupled from the benefits of submission. Authors can submit papers and benefit from others' reviewing labor without necessarily contributing proportionally themselves. While many conferences have reciprocal reviewing requirements, these primarily ensure quantity of reviews, but not quality.

#### 3.1.4. NO PUBLIC DATASETS FOR PEER REVIEW RESEARCH

The scientific study of peer review is severely hampered by the lack of public datasets. While reviews are made public in some conferences such as ICLR and journals like TMLR, various important pieces of data such as reviewer characteristics or bids is unavailable. Availability of such data can enable researchers at large to develop better peer-review processes. For instance, access to information about which set of reviews was done by the same reviewer can help research aiming to mitigate the problem of miscalibration (Mitliagkas et al., 2011; Freund et al., 2003; Wang & Shah, 2019). Access to information about attributes of reviewers can help develop better reviewer-paper assignment algorithms (Leyton-Brown et al., 2022). Data pertaining to bidding can help in developing better defenses against collusion rings (Jecmen et al., 2024). Without such datasets, it is far more challenging to make progress on these important problems.

### 3.2. The Proposed Reform: Anonymous and Non-anonymous Tracks

We propose that conferences offer two parallel tracks: one anonymous (as currently practiced) and one non-anonymous with full data release.

**The Anonymous Track.** The anonymous track operates exactly as conferences do today. Authors who prefer the

current system can choose to continue using it without modification. This ensures that the reform is opt-in and does not force participation from those with legitimate concerns about non-anonymous review. We expect that initially, most submissions would use the anonymous track, with the non-anonymous track serving as an experimental alternative that can demonstrate its value over time.

**The Non-Anonymous Track.** The non-anonymous track differs in several key ways:

**1. Author and reviewer identities hidden during review**: The process remains double-blind during the review phase, preserving the benefits of anonymity during evaluation. (The non-anonymous aspect applies only to the eventual release of data, not during the review process.) Reviewers evaluate papers without knowing author identities, and authors receive reviews without knowing reviewer identities, just as in the current system.

**2. Author reviewing commitment**: Each submitted paper must nominate at least one qualified author to review or meta-review other papers non-anonymously. This creates explicit reciprocity: if you want the community to review your work non-anonymously, and hence presumably with higher quality, then you commit to reviewing others' work non-anonymously as well.

**3. Full data release**: After the process, all data is released publicly, including bidding information, text-matching similarities, reviewer identities for each review, and meta-review content. The release should be done after an embargo period following the decision notification to the authors. A short embargo period (e.g., two weeks) would give authors time to thoughtfully process the decision by the time they can view the full review record.

**4. Review form for collegiality**: An important component of our proposal is a review form that promotes collegiality, an aspect that is vital for non-anonymous reviewing to succeed. Review forms that are currently employed, emphasize the 'accept vs. reject' decision. However, in non-anonymous review, publicly 'rejecting' the paper of another researcher (especially if the latter is senior) is challenging. Hence, in our proposal, we do not ask reviewers to make an accept versus reject decision. The review form contains only one textbox asking reviewers to summarize the paper, describe what they liked, and provide constructive feedback for improvement. Criticisms are framed as constructive feedback from a peer rather than as reasons for rejection. The final decision is made by meta-reviewers based on the totality of feedback. This design encourages reviewers to think of themselves as colleagues helping to improve work rather than as gatekeepers deciding who gets in. Indeed, an ideal feedback is from a colleague who is critical yet constructive – this is precisely what our pro-

posal aims to achieve.

Overall, we envisage that our design will encourage collegial feedback rather than adversarial judgment – something that is much needed in today's climate. Finally, the track is voluntary: authors uncomfortable with data release can use the anonymous track, so no one is forced into the non-anonymous system.

### 3.3. How This Reform Addresses the Problems

**Problem: No incentives for high-quality reviewing.**
*How it addresses this problem:* With public release, excellent reviewing would become visible and be rewarded through reputation effects. Reviewers who consistently provide thoughtful, constructive feedback build professional reputations for this contribution.

**Problem: No accountability for poor reviewing.**
*How it addresses this problem:* Reviewers cannot hide behind anonymity when providing low-effort or unprofessional reviews. The knowledge that reviews will eventually be public incentivizes care. Additionally, our proposed review form, which frames feedback as peer-to-peer suggestions rather than gatekeeping judgments, further encourages a collegial tone.

**Problem: Lack of reciprocity.**
*How it addresses this problem:* Our approach requires that authors wishing for non-anonymous reviews, which would presumably be of higher quality, should also provide non-anonymous reviews themselves. This requirement of reciprocity ensures that those who benefit from the system also contribute to it.

**Problem: No public datasets for peer review research.**
*How it addresses this problem:* The released data would enable rigorous study of peer review, supporting evidence-based improvements to the system for important problems, e.g., design and evaluation of better reviewer-calibration methods (Mitliagkas et al., 2011; Freund et al., 2003; Wang & Shah, 2019), better reviewer-assignment methods (Kobren et al., 2019; Payan & Zick, 2022; Singh et al., 2023; Stelmakh et al., 2025), models and statistical tests for bidding data (Cabanac & Preuss, 2013; Jecmen et al., 2024), commensuration biases (subjectivity) across reviewers (Noothigattu et al., 2021; Kitch & Shah, 2026), and others.

**Problem: Collusion Rings and Identity Theft**
*How it addresses this problem:* The non-anonymous track will have more consequences for attempts at fraud due to public nature of the reviewing information. The anonymous track is the same as current deployments.

# 4. Alternative Views

## 4.1. Against Separating Publication from Presentation

**Objection: Publishing all sound papers will lead to an explosion of low-quality publications.** *Response*: The low probability of presentation for unexciting work creates a natural disincentive for marginal submissions. Authors who care about visibility will still need to produce exciting work; those who submit sound but unexciting papers will receive publication credentials. Even today, a large amount of literature use happens based on arXiv submissions. But now we will have papers that are at least somewhat vetted for soundness. Furthermore, even though the literature may grow, search and recommendation systems can help researchers find relevant work. This is preferable to the current system where valuable work is rejected due to noise.

**Objection: Publication without selectivity devalues conference papers.** *Response*: The current apparent "signal" is considerably noisy, as demonstrated by the various experiments discussed earlier. A publication that signals "this work is technically sound" provides a more clear and meaningful signal than one that is based considerably on subjective assessments from a small number of reviewers, many of whom may have limited research experience. Additionally, selectivity for presentation still provides a signal, and this signal is based on broader community input rather than three potentially mismatched reviewers.

**Objection: Checking soundness is hard.** *Response*: We agree with this criticism. That said, we are simply following the objectives of peer review. To relieve some burden on human reviewers, one can employ human-AI collaborations. Such checking of soundness is already done in the Transactions on Machine Learning Research (TMLR) journal in our community. TMLR has criteria similar to our proposed Step 1, and in 2025 had an acceptance rate of about 69%, excluding desk rejected and withdrawn submissions (Kamath et al., 2025). A number of journals outside ML/AI also focus on soundness, such as the Public Library of Science (PLOS) ONE, Nature Scientific Reports, PeerJ, F1000Research, BioMed Central (BMC) Series journals, Royal Society Open Science, the Journal of Systems Research (JSys), Journal of Open Research Software (JORS), Journal of Open Source Software (JOSS), and many others.

Second, as discussed earlier, LLMs have been evaluated to be a useful tool in verifying soundness. The use of AI in Step 1 can thus significantly reduce the burden on the human reviewers.

**Objection: Checking soundness is not entirely objective.** *Response*: Yes, we agree that it is not 100% objective. For instance, there can be disagreements about the number or nature of datasets for any evaluation or the types of ab-

lation studies required to make a certain claim. That said, the degree of objectivity is significantly higher than criteria of interestingness and perceived impact.

**Objection: A correctness-first pipeline may inadvertently reject papers that are highly novel but imperfect, while favoring papers that are correct yet less impactful.** *Response:* This will not be an issue as long as the paper does not make false claims. To illustrate this, consider an example of a really novel paper which makes a false claim about beating a baseline. Suppose a reviewer catches this false claim. Then we do not want the false claim to be published. During the rebuttals, the paper can revise its claim to "this approach is novel but currently performs marginally worse than the baseline" which would be correct, retain novelty, and can now be accepted under our protocol.

**Objection: It should be the job of arXiv and not conferences to check for soundness.** *Response*: We do think conferences (and journals) should put efforts in ensuring soundness. They publish "peer reviewed" papers, which is generally considered by the public as well as by research evaluators as a signal of soundness. Furthermore, conferences have a large pool of reviewers who can participate in reviewing whereas arXiv does not.

**Objection: The separate evaluation of subjective aspects can still have correlated biases.** *Response*: We agree that while the variance may go down with more samples, biases correlated across reviewers (e.g., due to promotion of the paper on social media by authors) may continue to exist. That said, the much larger reviewer pool in Step 3 makes it harder for any single source of bias, such as one viral tweet, to dominate the aggregate signal. We also note that correlated biases exist in the current system as well.

## 4.2. Against Non-Anonymous Tracks

**Objection: Non-anonymous reviewing will lead to retaliation and career harm, especially for junior reviewers.** *Response*: The non-anonymous track is voluntary. Furthermore, the simplified review form removes accept/reject recommendations, and reframes criticism as constructive feedback. A review that says "here is how to strengthen the experimental methodology" is less likely to generate retaliation than one that says "reject due to weak experiments."

**Objection: Public data release raises privacy concerns.** *Response*: We understand these concerns, which is why the proposed non-anonymous track is voluntary. Researchers who value privacy can submit to the anonymous track, and those who participate in the non-anonymous track do so with full knowledge that their reviewing information will be public. Moreover, other research communities have successfully adopted non-anonymous peer review (see, for in-

stance, `http://f1000research.com/`).

**Objection: Only papers with senior authors who can also review will submit to the non-anonymous track.** *Response*: First, if many senior researchers review, that will be a good thing for peer review at our conferences. Second, we agree that initially there may be more hesitancy from junior researchers. However, we think time is ripe to try out such a reform. Over time, we hope that even junior researchers will see the benefits of this approach and participate more in it. In the future, one could make additional accommodations to facilitate this, such as giving more weight to junior researchers' bids in the reviewer assignments. Third, those who follow discussions of ML/AI research in social media may also have seen junior researchers frequently point out problems in papers authored by senior researchers. Fourth, some reviewers who generally put significant effort in their reviews may even be incentivized to sign up for the non-anonymous track to receive due credit for their efforts. This makes us more optimistic about the early uptake of the proposed reform.

**Objection: Fraud such as collusion rings and identity theft will continue to prevail.** *Response*: As discussed in earlier sections, we believe that collusion ring and identity theft will be mitigated by our four step process as well as in non-anonymous reviewing. That said, we do not claim it will entirely stop such attacks. Attempts of fraud are likely to continue, and could still succeed.

## 5. Call to Action

The current peer-review system is showing serious limitations, and addressing them will require significant reforms rather than minor adjustments. We call on the machine learning community to take concrete steps toward trying out these reforms. The deployments can be done at a standalone conference, and do not require coordination across conferences.

We urge program chairs and conference boards to pilot the four-step process (even for a subset of submissions). TMLR's journal-to-conference track and NLP's "findings" tracks implement some aspects of this proposal. As a starting point, these partial solutions can be extended into a broader framework. A natural starting point is to expand the TMLR journal-to-conference track or to create a parallel "soundness-first" track that implements Steps 1 and 2 of our proposal. Data from these pilots can be analyzed to evaluate the reform's effects. In a similar vein, we also urge for the introduction of a non-anonymous track as an experimental option, allowing authors to opt in while gathering data on outcomes. Even a small non-anonymous track (e.g., 100 papers) would generate valuable data about how transparency affects review quality.

More broadly, we also urge the community to support and participate in novel attempts at reforming peer review. There will inevitably be hiccups along the way, but our community which prides itself on innovation and evidence-based reasoning should also be willing to experiment with its own institutions. We cannot know what works without trying, and we cannot improve without tolerating some initial failures.

## 6. Conclusions

The peer-review process in ML/AI conferences needs major reforms. We are lucky to be a part of a research community that has done numerous experiments measuring various aspects of the review process. Leveraging these experiments, this position paper argues for two policy shifts, in an evidence-based fashion. Subsequent to deployments, the proposed policies can be evaluated using similar tools as the experiments previously conducted in the current ML/AI conference review process. We urge the community to continue taking a scientific approach to the design of scientific peer review.

When the ICLR conference was launched, it took the novel approach of publicly releasing rejected papers and also all reviews. When TMLR was launched, it took the approach of evaluating only correctness. Now after several years, both ICLR and TMLR are very well regarded in the community. There is thus significant precedence of novel review protocols in our field, and its acceptance and adoption by the community.

Our final remarks discuss the impact on evaluators of researchers such as promotion, hiring and admissions committees. For Reform 1, the starting point for such evaluators is acknowledging that the current system is failing at both of its stated objectives: it neither reliably catches flaws nor necessarily identifies stronger work. This problem will likely worsen with increasing AI-fueled submissions. Evaluation systems should recognize that a soundness-first process emphasizes correctness, and that community-based presentation selection draws on more opinions than the current three-reviewer panels thereby reducing variance in quality judgments. For Reform 2, evaluators may incorporate the quality of reviewing into assessments of researchers. A prior effort in this direction, Publons, made review counts publicly visible, but its focus on quantity rather than quality limited its usefulness. Our non-anonymous track addresses this directly because the reviews themselves are public and evaluation can be based on the reviews themselves rather than merely a count of the reviews. Our goal is to provide a higher signal-to-noise ratio in research evaluation, and we envisage committees will adapt appropriately to new modalities and protocols.

## Acknowledgments

This work was supported by grants NSF 1942124 and ONR N000142512346.

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
