# OpenReview forum: "Position: Peer Review in ML/AI Conferences Should Separate Publication from Presentation and Offer Non-Anonymous Review Tracks"
_ICML.cc/2026/Position_Paper_Track — ICML 2026 Position Paper Track regular_

### Official Review · Reviewer_KPWH · 2026-02-26

**Significance:** 3
**Argument Clarity:** 3
**Rating:** 4
**Confidence:** 3

**Questions:**

1.How to ensure that the reviewers in the first step have sufficient willingness and ability to conduct in-depth verification?

2.How to design a presentation selection strategy to ensure fairness?

3.How to attract researchers to participate in the non-anonymous review track in the early stage?

**Alternative Views Section:**

Yes

**Compliance With Llm Reviewing Policy A Conservative:**

Affirmed.

**Discussion Potential:**

3

**Final Justification:**

The rebuttal addressed my main concerns,  and I agree with the acceptance of this paper.

**Paper Summary:**

This paper is a well-structured position paper with important practical significance. Based on some evidence in previous conferences, such as NeurIPS 2014, the authors indicate some issues within the current peer review in ML/AI conferences. Subsequently, two specific proposals are proposed correspondingly, i.e., (1) separating publication from presentation via a four-step process, and (2) offering parallel anonymous and non-anonymous reviewtracks. The paper not only lists the shortcomings of the current status but also constructs a comprehensive solution framework, rigorously addressing potential counterarguments. Therefore, it may promote further discussions within the community regarding the reform of the review system.

**Position:**

Yes

**Position In Title:**

Yes

**Related Work:**

3

**Strengths And Weaknesses:**

Strengths:

1.For both reforms, the authors utilize real examples to support the problems they indicate. For instance, the experiments in Shah, 2022 and Table 1 are to support the failure to ensure correctness, and the experiments in NeurIPS 2014 are to support the failure to filter reliably.

2.Separating publication from presentation is a novel paradigm in which researchers can select the most interesting papers by themselves.

3.The non-anonymous review track can improve peer reviews.

Weaknesses:

1.In the Four-Step Process, the first step Correctness Evaluation imposes more strict requirements on reviewers from the technical rigor and topical relevance perspectives. However, the subjective assessments of “importance” are also important for a paper. Without significant importance, this paper is meaningless, no matter how technically correct it is.

2.Although the third step Community Excitingness Rating allow the researchers to select papers, the difference in the counts of researchers among different fields may make negative impact, e.g., most presentations may belong to a few popular fields, hindering the development of other fields.

3.Please explain how to resolve the resubmission problem in detail.

4.The paper points out that the two reforms can be implemented independently, but it does not delve into the potential complexities that may arise from their combination. For example, in the "first step", can we adopt "non-anonymous review" to further improve the quality of correctness checks? In the "third step", will disclosing reviewers’ identities affect the scoring?

**Support:**

3

---

> ### Author Rebuttal · Authors · 2026-03-29
>
> Thank you very much for your review of the paper and the four questions.
>
> Q1. Even the current review process asks reviewers to check for correctness (along with interestingness), and the well regarded journal Transactions on Machine Learning Research (TMLR) focuses only on correctness.  So there is already precedent for this. Our approach seeks to improve the current state in conferences. First, as discussed in the paper, it is shown by Lane et al. (2024) that asking for multiple evaluation criteria simultaneously leads to poorer evaluation along individual criteria rather than asking individual criteria separately. We address this in our proposal. Second, we envisage the correctness evaluation to be done by a combination of human plus LLM (in a manner that program chairs see fit). It is found in prior works (e.g., Liu et al. 2023) asking LLMs to specifically identify flaws leads to a higher signal as compared to asking it to write a general review. Third, current conferences ask reviewers to review about 6 papers which is a large workload. We suggest a much lower workload in the first stage of about 2 papers per reviewer, which would allow reviewers to focus more on each paper and should suffice (along with LLMs) for a reasonably high per-paper signal.
>
> Q2. Our current approach is similar to what is done today, where today three reviewers provide their subjective opinions on interestingness, and based on this papers are selected for presentation at the conference. But importantly, under our proposed approach, this will be done with a much bigger sample size thereby improving the signal to noise ratio.
>
> Q3. It appears that your sentence got cut off halfway. We discuss resubmissions in W3 below.
>
> Q4. We expect that initially, there will be grater participation in the non-anonymous stage from more senior researchers. That said, many junior researchers openly post critiques of papers publicly on social media, so one may envisage that some junior researchers may also join. Going ahead, with the safeguards we have put in place in our design (e.g., not having to make an acceptance/rejection decision), we hope that more junior researchers will begin seeing the benefits of joining the non-anonymous track.
>
>
> W1. You are right that importance is also important. However, our argument is that the current evaluation of importance is extremely noisy, and so is the evaluation of correctness, and current processes further conflate the two. The proposal will help obtain a stronger signal on this measurement, which at the end of the day, is still subjective.
>
> W2. This is a challenge in today’s conference as well, if reviewers in one field are generally more positive than reviewers in another field. We assume the same methods that are used in current conferences will apply there, where calibration across papers and fields is achieved via SACs and program chairs.
>
> W3. Currently, 75% to 80% of submissions to conferences are rejected, frequently due to subjective comments by reviewers. A large fraction of these submission are resubmitted to a subsequent conference, thereby increasing the reviewing volume. The proposed approach would nip this in the bud -- if the paper is deemed correct then it is accepted at the end of the correctness stage (irrespective of the subjective evaluation in the interestingness stage) and cannot be resubmitted. Thus this large volume of resubmissions will be mitigated.
>
> W4. If implementing both together, then yes, it would be non anonymous in the correctness stage so that reviewers can provide objective comments to authors. That said, if conferences are interested in these approaches, we would recommend them to go slow and implement them one at a time, observe the outcomes, and then combine the two.

---

> > ### Author Rebuttal · Reviewer_KPWH · 2026-04-01
> >
> > Thanks for the detailed response. I will hold the rating.

---

### Official Review · Reviewer_vq19 · 2026-03-01

**Significance:** 3
**Argument Clarity:** 3
**Rating:** 4
**Confidence:** 3

**Questions:**

Please see the comments above. Also:

1. Can we estimate how many submissions in ICLR-scale conferences pass the correctness criteria? What are the implications towards practical value of the proposal and reviewer capacity?

2. How do you anticipate hiring and promotion committees adapting to correctness-only publication signals? Could volume inflation undermine the reform?

**Alternative Views Section:**

Yes

**Compliance With Llm Reviewing Policy A Conservative:**

Affirmed.

**Discussion Potential:**

3

**Final Justification:**

My main concerns center on feasibility and incentive design. In particular, the practicality of correctness-first evaluation at ICML/NeurIPS scale remains unclear, especially given the lack of estimates of how many submissions would pass such a filter and the associated reviewer capacity required. The rebuttal acknowledges this uncertainty and provides a rough reference point using TMLR, but this does not fully address scalability at conference scale.

I also remain unconvinced that the proposal must be implemented at the conference level. While the rebuttal clarifies that conferences provide the necessary infrastructure for adjudication and appeals, it does not fully justify why correctness-first publication could not be handled by alternative dissemination models (e.g., journal-style or arXiv-adjacent systems) with lighter-weight governance.

The rebuttal clarifies several aspects of the proposal and improves its positioning, but it does not materially change my assessment of these core feasibility and incentive concerns.

**Paper Summary:**

The submission argues that ML/AI conference peer reviews currently fail both core objectives of peer review: ensuring correctness and reliably filtering for impactful work. The authors perform controlled error detection experiments, reviewer consistency studies, and citation prediction analysis, and argue that conferences frequently miss technical flaws, exhibit inconsistency, and fail to predict impact reliably. The submission proposes two reforms:

Reform 1: A process seperating correctness evaluation from publication and presentation. Technically sounds papers are published first, then presentation slots are allocated based on community excitement's.

Reform 2: Parallel anonymous and non-anonymous review tracks.

**Position:**

Yes

**Position In Title:**

Yes

**Related Work:**

3

**Strengths And Weaknesses:**

Strength:

1. The paper performs a structural systematic diagnosis considering various factors such as misaligned incentives.

2. The diagnosis draws from several studies, including error detection experiments, consistency replication, score - citation analysis, etc.

3. The discussion potential of the proposals seem high and the call to action is concrete.

Weakness:

1. Scaling correctness checking to large conferences can be demanding. LLM assistance may remain partial.

2. It's not completely clear if the presentation selection may change incentives. Many institutions are already differentiating between, say short paper vs long papers. If technical correctness is a filter, it's not clear why it should be done at conference level, instead of, say, arxiv.

3. Community excitement can be subjective with many confounding factors, and gaming of the system.

**Support:**

3

---

> ### Author Rebuttal · Authors · 2026-03-29
>
> We thank the reviewer for their time and effort in reviewing the paper.
>
> Q1. We do not know how many submissions in large conferences pass the correctness criteria. It is not clear how to perform this estimation in a rigorous manner without any ground truth. If one were to simply compute the fraction of cases the reviewers claim an error or an LLM claims an error, we will be making strong assumptions on the accuracy of human or LLM reviewing.
>
> That said, we can use The Transactions on Machine Learning Research (TMLR), whose criteria are very similar to our correctness criterion, to get an idea. This report from the TMLR website says that they had a 70% acceptance rate.
> https://docs.google.com/document/d/1EsApC9y08bY2Qv2lQ-1NjTgU8QuAT9E30k44QiuSG5c/edit?tab=t.0
>
> Q2. We will answer this question in two parts.
> a) The current evaluation of conferences and use in hiring/promotion committees is anyways misaligned. These committees frequently put far higher emphasis on acceptances in these conferences, even though we have a number of experiments showing that the signal is actually very weak. The proposed scheme aims to improve the signal to noise ratio by first focusing on correctness (which is a prerequisite for good science) and then having more samples on subjective aspects.
> b) TMLR has now existed for at least 3  years and is regarded well by the community. Committees have adapted to papers published in TMLR, and likewise they can adapt to other signals like the correctness and interestingness stage we propose.
>
>
>
> W1. We envisage human+LLM evaluation of correctness. Prior work as discussed in the paper has found that LLMs are better at checking correctness than merely “writing a review.”
>
> W2. There is a key difference between arXiv and conferences which makes conferences (or journals) more suitable for doing such evaluations. Conferences have a large pool of reviewers who can perform evaluations, or even if LLMs do evaluations, humans can adjudicate LLM outputs and any appeals. ArXiv does not have that infrastructure since its entire model of operation is quite different.
>
> W3. That is true in current processes as well. In fact, in current processes it is *worse* since only 3 reviewers provide their excitement scores for a paper. Our proposal will reduce this problem with a higher sample size in the interestingness step.

---

> > ### Author Rebuttal · Reviewer_vq19 · 2026-04-02
> >
> > Thank you for the detailed responses and clarifications.
> >
> > On the “conference vs arXiv” distinction: I appreciate the clarification that conferences provide the infrastructure (reviewers, adjudication, appeals) necessary to support correctness evaluation. This helps clarify the intended scope of the proposal. However, I still find the justification somewhat incomplete, as it remains unclear why correctness-first publication must be coupled to conference processes rather than being handled by alternative dissemination models with lighter-weight governance.
> >
> > On correctness-first feasibility: The discussion and reference to TMLR are helpful, but the feasibility at ICML/NeurIPS scale remains uncertain. In particular, the lack of an estimate of how many submissions would pass correctness, and the associated reviewer capacity required, continues to be a central open question.
> >
> > On community-based excitingness: I understand the argument that increasing the number of raters may reduce variance. However, concerns about correlated bias (e.g., popularity effects or visibility-driven feedback loops) are not fully addressed.
> >
> > On non-anonymous review track: The clarification of a gradual rollout is helpful. However, the proposal still relies largely on anticipated behavioral changes (e.g., participation by senior/junior researchers) without supporting evidence from comparable systems.
> >
> > Overall, the rebuttal clarifies several aspects of the proposal but does not materially change my assessment of its feasibility and incentive implications. Keeping the rating for now and will be keen to look at other reviewers' responses.

---

### Official Review · Reviewer_HfVC · 2026-03-05

**Significance:** 3
**Argument Clarity:** 3
**Rating:** 5
**Confidence:** 3

**Questions:**

1. For the 4 step process, maybe I'm missing something but, each step will have differente reviewers?
2. What is your opinion regarding the novelty and the impact of the poor quality papers that can be accepted in the first step?

**Alternative Views Section:**

Yes

**Compliance With Llm Reviewing Policy A Conservative:**

Affirmed.

**Discussion Potential:**

4

**Final Justification:**

Concerns were fully addressed.

**Paper Summary:**

In this paper, the authors present two important reforms to the process of publication and presentation at ML/AI conferences. The first reform consists of separating publication from presentation. They explicitly emphasize the difficulty in verifying the accuracy of a previously published article, as well as the inconsistencies among review panels. Another important point for this first reform is the considerable increase in revisions due to resubmissions. Finally, the authors propose a four-part process for this first reform.
The second reform involves separating the main track of each conference into two tracks: one anonymous and the other public, where the community can voluntarily submit papers. The authors justify this reform by explaining that there are currently no incentives for high-quality peer review. Specifically, the anonymous track would function exactly as it has until now, while the non-anonymous track would be experimental, as it is likely that few authors will prefer this new format.

**Position:**

Yes

**Position In Title:**

Yes

**Related Work:**

3

**Strengths And Weaknesses:**

The paper is well-written, and the authors' arguments supporting their position are well-formulated. They present some statistical data to bolster their stance on a problem that has been growing in the community in recent years. It is a highly debatable topic, ranging from objective to subjective perspectives, where individuals may take a particular stance based on their own experience and that of their colleagues.

**Support:**

3

---

> ### Author Rebuttal · Authors · 2026-03-29
>
> We thank the reviewer for their careful reading of the paper.
>
> Q1. Step 4 can reuse the reviewers in the correctness step. This is because the reviewer has already read the paper. That said, this step will obtain a number of additional samples. We will clarify this in the revision.
>
> Q2. To understand this, the Transactions on Machine Learning Research (TMLR) is a good building block. As discussed in Section 2.2, our first step criteria is similar to that employed by TMLR. The impression of a TMLR paper is that it is a technically solid paper which has been checked more rigorously than a NeurIPS/ICML/ICLR paper.
>
> As also discussed in the paper, it is also not clear how well current ICML/NeurIPS/ICLR capture novelty. For instance, even the results in ICML best paper awardee was alleged to have been known since at least ten years. The argument we are making is that the focus on correctness first and then more samples on interestingness will help improve the signal to noise ratio.

---

> > ### Author Rebuttal · Reviewer_HfVC · 2026-04-02
> >
> > Thank you for the response and for address my concerns. I will rise my score.

---

### Official Review · Reviewer_oXUn · 2026-03-12

**Significance:** 3
**Argument Clarity:** 4
**Ethics Flag:** Yes
**Rating:** 5
**Confidence:** 3

**Questions:**

Q1. Could a correctness-first pipeline inadvertently reject papers that are highly novel but imperfect, while favoring papers that are correct yet less impactful? If so, how would the framework avoid slowing progress in the field?

Q2. Can community-rated “excitement” reliably reflect research quality and significance? Visibility can be shaped by publicity and group influence, which may favor popular papers over truly important ones and reduce diversity in selected presentations.

Q3. Since multiple channels already track paper attention (e.g., community trend platforms), how would this additional excitement rating avoid redundancy? What governance would make it authoritative, who should provide ratings, and how can participation be encouraged while maintaining fairness?

Q4. What changes in academic and industry evaluation systems are needed to better recognize reviewer contributions and make this proposal practically adoptable?

Q5. The proposed strategy seems to require cross-conference coordination. Could the authors outline a rollout plan—for example, piloting at major venues such as NeurIPS/ICML/ICLR before broader adoption across other conferences and communities?

**Alternative Views Section:**

Yes

**Compliance With Llm Reviewing Policy A Conservative:**

Affirmed.

**Discussion Potential:**

4

**Final Justification:**

The authors have addressed my concerns. Hence, I raise my rating to Accept.

**Paper Summary:**

This paper argues that peer review in current ML and AI conferences is systematically weak on two core goals: verifying technical correctness and reliably filtering for high-quality work. Drawing on empirical evidence, the authors highlight frequent failure to detect major flaws, high decision inconsistency across reviewer panels, reviewer overload from resubmissions, weak incentives for review quality, and limited transparency and data for process improvement. They propose two concrete reforms: a four-step pipeline that separates publication from presentation, with correctness assessed first and community ratings then used to select talks, and parallel anonymous and non-anonymous review tracks to improve accountability and create public datasets for peer review research. The paper’s central position is that structural process redesign, rather than minor tuning, is needed to make peer review more reliable, efficient, and scientifically useful.

**Position:**

Yes

**Position In Title:**

Yes

**Related Work:**

3

**Strengths And Weaknesses:**

**Strengths**

S1. The paper clearly identifies two core goals of peer review, ensuring rigor and filtering for quality, and builds its proposal around this framing. Its four-step process is well motivated and could improve both author and reviewer experience while offering constructive direction for the broader community.

S2. The position advanced in this paper is highly interesting and is likely to stimulate substantial discussion in the community.

S3. The discussion of alternative views is comprehensive and detailed, and it effectively clarifies points that are often conflated or misunderstood, which strengthens the paper’s position.

S4. The paper is well organized and easy to follow.

**Weaknesses**

W1. The evidence in Sec. 2.1 is somewhat limited in both recency and scale. Given the rapid growth of AI conferences and the increasing use of LLMs in writing and reviewing, the empirical support would be stronger with broader and more up-to-date analyses.

W2. Separating publication from presentation may work well for conference papers, but the proposal appears less applicable to AI/ML work published in journals.

**Support:**

4

---

> ### Author Rebuttal · Authors · 2026-03-29
>
> We thank the reviewer for their thoughtful remarks and questions.
>
> Q1. That won’t be an issue as long as the paper does not make false claims. To illustrate this, let us consider an example of the workflow. Suppose there is a really novel paper but it makes a false claim about beating a baseline. Suppose a reviewer catches this false claim. Then we don’t want the false claim to be published. So after the rebuttals, the paper can revise its claim to “this approach is novel but currently performs marginally worse than the baseline” which is correct, retains novelty, and now can be accepted under our protocol.
>
> Q2. The current process has papers’ quality and significance rated by just *three* people. The proposal will increase it to many more due to the light reviewing load in the second round. In this manner, we envisage improving upon current practices via the proposed approach.
>
> Q3. The assignment of raters in the second round will be the same as the reviewer assignment process done in OpenReview today. The difference is a much bigger sample size per paper.
>
> Q4. When ICLR started, they began the novel approach of publicly releasing rejected papers and also all reviews. Committees adapted to consider this signal. When TMLR started, they began the approach of evaluating only correctness. Again, committees adapted to this signal. Now both ICLR and TMLR are doing well. Our goal is to provide them with a higher SNR in research evaluation, and we envisage committees to adapt appropriately to new modalities and protocols.
>
> That said, since the reviewer asked, we share concrete changes that can be made. For Reform 1, the starting point is acknowledging that the current system is failing at both of its stated objectives: it neither reliably catches flaws nor identifies stronger work. This problem will likely worsen with increasing AI-generated submissions. Evaluation systems should recognize that a correctness-first process emphasizes rigor, and that community-based presentation selection draws on more opinions than the current three-reviewer panels thereby reducing variance in quality judgments. For Reform 2, evaluators may incorporate the quality of reviewing into assessments of researchers. A prior effort in this direction, Publons, made review counts publicly visible, but its focus on quantity rather than quality limited its usefulness. Our non-anonymous track addresses this directly because the reviews themselves are public and evaluation can be based on what a reviewer actually wrote.
>
> Q5. We do not posit nor require cross-conference coordination. Just like ICLR started open reviewing on its own, this proposal can be adopted by an individual conference.
>
>
> W1: These are all the experiments we know of that have been done in active reviewing processes. There are separate experiments such as the ICLR reviewer assistance experiment and the AAAI AI reviewer experiment, but none of these have ground truth and hence cannot measure the rates of errors being caught. Note that we do discuss LLM reviewer’s ability to find errors in Section 2.2 (paragraph starting with “The evaluation in this stage can combine human reviewers with LLM-assisted checking.“).
>
> If the reviewer is aware of additional experiments objectively quantifying whether current review processes can meet the objectives of reviewing, we would be glad to also include them.
>
> W2: Yes, the reviewer is correct. The current proposal focuses on conferences such as ICML, NeurIPS, ICLR, AAAI etc. which combine presentation and publication into one evaluation.

---

> > ### Author Rebuttal · Reviewer_oXUn · 2026-04-03
> >
> > Thanks for your response. I will raise my rating based on your response and the opinions from other reviewers. Please revise the paper accordingly. Specifically,
> >
> > (1) Please make clear that novel yet imperfect papers with correct claims will not be missed.
> >
> > (2) Please make clear that, in the second stage, the so-called community refers to qualified reviewers, not everyone on social media.
> >
> > (3) Please make clear that the proposed reform can act in a stand-alone conference.
> >
> > I wish that a clearer statement of the reform can convince the audience that it is practical and will bring visible benefits.

---

### Decision · Program_Chairs · 2026-04-30

**Decision:**

Accept (regular)

**Comment:**

All reviewers liked the paper.  Well structured and presented.
Lots of questions about the proposal, but none were against.  Some questions around feasability.
Should generate good discussions.